# Geographical barriers and multimorbidity in quilombola territories of the amazon region

**Leanna Silva Aquino**[1☯], **Ellen Mara Fernandes da Silva**[1‡], **Victoria Valentim Aguiar**[2‡],
**Cesar Ferreira Fernandes Filho**[3‡], **Sheyla Mara Silva de Oliveira**[1‡], **Tatiane Costa Quaresma**[1‡],
**Valney Mara Gomes Conde**[1‡], **Nádia Vicência do Nascimento Martins**[1‡],
**Marcos Manoel Honorato**[1‡], **Veridiana Barreto do Nascimento**[1‡],
**Guilherme Augusto Barros Conde**[4‡], **Franciane de Paula Fernandes**[1‡],
**Lívia de Aguiar Valentim**[1☯]*

1 Department of Health, Center for Biological and Health Sciences, Campus XII Section, State University of Pará, Santarém, Pará, Brazil, 2 Department of Health, Metropolitan College of Pará – FAMETRO, Santarém, Pará, Brasil, 3 Department of Health, University of Amazonia – UNAMA, Santarém, Pará, Brasil, 4 Department of Biosciences, Federal University of West Para- UFOPA, Santarém, Pará, Brasil

☯ These authors contributed equally to this work.
‡ EMFDS, VVA, CFFF, SMSDO, TCQ, VMGC, NVDNM, MMH, VBDN, GABC AND FDPF also contributed equally to this work.
* livia.valentim@uepa.br

## Abstract

### Background

Quilombola communities in the Brazilian Amazon face persistent social and territorial inequities that shape health outcomes and access to care. Geographic isolation, limited transportation, centralization of specialized services, and socioeconomic disadvantages contribute to unequal opportunities for timely diagnosis and treatment. Understanding how these determinants interact with patterns of multimorbidity is essential for guiding equiTable health policies and strengthening primary care in remote territories.

### Methods

A cross-sectional epidemiological study was conducted with 518 adults from nine quilombola communities in Santarém, Pará. Data were collected through household surveys addressing sociodemographics, self-reported diseases, service utilization and resolvability. Geographic coordinates of communities and health services were mapped to classify accessibility as high, medium or low. Diseases were converted into a binary matrix to estimate prevalence and identify multimorbidity (≥2 conditions). Statistical analyses included chi-square tests, ANOVA, Spearman correlations and heatmap visualization. A Composite Access Index (CAI) integrating geographic distance, epidemiological burden and service-use indicators was developed. A Random Forest model was used to identify conditions most strongly associated with multimorbidity.

**Data availability statement:** The minimal dataset analyzed in this study is not publicly available due to ethical and privacy restrictions. The dataset contains sensitive patient information and variables that, even after anonymization, could allow re-identification; therefore, open sharing was restricted by the Research Ethics Committee of the Universidade do Estado do Pará (UEPA) (CEP Tapajós). In accordance with the PLOS Data Availability Policy, access to the minimal dataset may be granted upon request submitted directly to the institutional contact point—CEP Tapajós (UEPA): Av. Plácido de Castro, 1399, Bairro Aparecida, Bloco I, sala 02, Santarém–PA, CEP 68.040-090, Brazil. Phone: +55 (93) 3512-8013 | Fax: +55 (93) 3512-8000, subject to ethics review, execution of a confidentiality/data-use agreement when applicable, and compliance with applicable data protection regulations.

**Funding:** The author(s) received no specific funding for this work.

**Competing interests:** The authors have declared that no competing interests exist.

**Abbreviations:** ANOVA, Analysis of Variance; CAAE, Certificate of Ethical Approval Submission; CAI, Composite Access Index; CAPS AD III, Psychosocial Care Center for Alcohol and Drugs; CAPSi, Child Psychosocial Care Center; CEP, Research Ethics Committee; CEREST, Occupational Health Reference Center; CRS, Specialized Reference Center (e.g., Elderly, Women, Child); CTA/SAE, Counseling and Testing Center/ Specialized Assistance Service; PHC, Primary Health Care; SDH, Social Determinants of Health; SAMU 192, Mobile Emergency Care Service; SUS, Unified Health System; UBS, Basic Health Unit; WHO, World Health Organization.

## Results

Communities showed marked territorial heterogeneity. Pérola do Maicá had the highest accessibility, while Ituqui, Tiningu and Murumuru presented substantial geographic and logistical barriers. Service utilization ranged from 42.9% to 95.0%, and most communities relied on care outside their territory (70–95%). Complete problem resolution was reported by 72.5% of participants, though with variation among communities. The CAI identified Ituqui (0.550), Tiningu (0.480) and Murumurutuba (0.331) as the most vulnerable territories. The Random Forest model achieved 93.6% accuracy, with hypertension, diabetes, musculoskeletal diseases, arthritis/rheumatism and heart disease emerging as key predictors of multimorbidity.

## Discussion

Findings indicate that social and territorial determinants are strongly associated with inequities in access to health services, continuity of care, and disease burden across quilombola communities.

## Conclusions

Geographic barriers and the distribution of health services are associated with distinct patterns of multimorbidity and health service access among quilombola populations. Strengthening primary care, transportation, and diagnostic support may help mitigate inequities and improve health conditions in remote Amazonian territories.

## Introduction

Quilombola communities in the Amazon region of Pará have experienced, for decades, a set of social inequalities that manifest directly in the health-disease process and in access to health services. These inequalities are rooted in historical, political, and economic conditions that shape the distribution of opportunities across the territory, producing persistent scenarios of vulnerability. The literature shows that factors such as income, education, occupation, environmental conditions and territorial characteristics strongly influence the possibilities of care, positioning the social determinants of health (SDH) as a central framework for understanding inequities among different population groups [1–3].

In the Brazilian context, SDH have been recognized as factors that determine how individuals and communities become ill, remain healthy and access services provided by the Unified Health System (SUS). Models such as Dahlgren and Whitehead's highlight that structural elements, including public policies, living and working conditions, and socioeconomic factors, interact with individual characteristics and lifestyles to produce avoidable and unjust inequalities [4]. Among traditional populations, such as quilombolas, these layers of determination become even more complex, as geographic barriers, territorial isolation, limited service availability and structural constraints overlap, reinforcing cycles of illness and reducing the resolvability of care [5–7].

The theory of health inequities proposed by Michael Marmot further deepens this analysis by demonstrating how historically constructed social inequalities reverberate in measurable differences in health outcomes. According to the author, material conditions, socioeconomic position and the social meanings attributed to these conditions shape trajectories of illness and well-being. Recognizing how such inequalities are distributed across territories and among marginalized populations is therefore essential [2,3]. In quilombola communities, this perspective reveals that insufficient access to essential goods and services, such as education, income and sanitation infrastructure represents not only material scarcity but also rights violations with direct impacts on daily life.

Recent studies in quilombola territories have shown that difficulties in transportation, long distances to health units, absence of regular transit, limited access to river-based emergency services and dependence on privately owned boats in emergencies constitute a reality that directly affects the ability to seek timely care [7–9]. These constraints are compounded by socioeconomic restrictions, underdiagnosis and challenges in continuity of care, reinforcing that territory is not merely a backdrop but an active determinant of health.

Understanding the social determinants that influence access to health care in the quilombola communities of Santarém is therefore essential to identify structural inequalities, recognize patterns of vulnerability and inform more equiTable public policies. An integrated analysis of these factors expands the understanding of how geographic, socioeconomic, cultural and organizational barriers shape the use of health services and their resolvability. Additionally, it highlights the urgency of strategies that strengthen Primary Health Care (PHC), expand service coverage and promote the right to health in its fullness, considering the specificities of traditional populations in the Amazon.

## Methods

This study is characterized as an epidemiological, cross-sectional investigation with a quantitative approach, conducted to analyze geographic, organizational, and care-related accessibility to health services among quilombola communities in the municipality of Santarém, Pará, as well as to identify the distribution and patterns of self-reported diseases and factors associated with multimorbidity. The research was carried out in nine communities of the Lower Amazon region, Arapemã, Saracura, Pérola do Maicá, Bom Jardim, João Pereira, Murumurutuba, Murumuru, Tiningu, and Ituqui, which present territorial heterogeneity marked by differences in distance from the municipal center, distinct environmental conditions, and modes of transportation influenced by river seasonality. These regional variations reflect diverse possibilities for accessing health services, justifying the choice of territory as the central axis of the analysis.

The sample definition was based on population estimates from the 2022 Demographic Census conducted by the Brazilian Institute of Geography and Statistics (IBGE), which identified 4,363 individuals who self-declared as quilombolas in the municipality of Santarém, of whom 2,133 resided in officially recognized quilombola territories. Based on this population, the sample size was calculated assuming a 95% confidence level and a 5% margin of error, resulting in a minimum estimated sample of approximately 326 participants. Participant selection followed a non-probabilistic consecutive sampling approach, combining territorial and institutional recruitment strategies.

In each community, all quilombola residents aged 18 years or older, present at the household at the time of data collection and who voluntarily agreed to participate, were considered eligible. Individuals with cognitive limitations that impaired comprehension of the questionnaire or communication of responses, as well as those absent after up to two contact attempts, were excluded. Recruitment occurred in four main contexts: (1) at the Basic Health Unit serving as the reference facility for the quilombola population of the Tiningu community; (2) during meetings of community associations, where the study was presented to local leaders and families; (3) during household visits; and (4) during itinerant health actions conducted in partnership with the Omulu Project. Refusals were infrequent and were mainly related to lack of availability at the time of approach or prolonged absence from the household, with no significant concentration of non-responses in specific communities. At the end of the recruitment process, 518 individuals participated in the study, exceeding the initially estimated sample size and thereby strengthening the representativeness of the sample.

Data collection was conducted by trained researchers, ensuring greater accuracy of responses and adequate territorial understanding. The questionnaire included sociodemographic information, identification of up to 23 self-reported diseases, history of health service utilization, care resolvability, and characteristics related to mobility, transportation, and access difficulties.

Health conditions were assessed through self-report using a structured questionnaire administered by trained interviewers. Participants were asked whether they had ever been informed by a health professional (physician or nurse) that they had specific chronic or acute conditions, based on a predefined list of diseases. Whenever necessary, interviewers used examples of commonly prescribed medications, previous exams, or follow-up care to facilitate understanding and improve recall. The questionnaire did not assess clinical symptoms in the absence of a prior diagnosis, and conditions were recorded only when participants explicitly recognized a previous health professional diagnosis. Self-reported morbidity was therefore used as a proxy for diagnosed conditions rather than current symptoms. Specific variables such as "care in the last year," "location of care," "problem resolution," and "health conditions" were used as key indicators for analyzing access and service performance.

After data collection, variables were standardized and statistically processed. Self-reported diseases were separated and converted into a binary matrix, allowing the calculation of prevalence by community and the identification of individuals with multimorbidity, defined as the presence of two or more simultaneous diseases. Geographic coordinates of communities and health services were obtained through latitude and longitude, enabling the construction of georeferenced maps representing territorial distribution and distance gradients.

Some analyses were conducted using specific subsamples of participants, depending on the research question and data availability. In particular, indicators related to health service resolvability were calculated only among individuals who reported having accessed health services in the 12 months preceding the interview. Therefore, the denominators (N) vary across tables and analyses and are explicitly indicated for each result to ensure clarity and avoid misinterpretation.

All cartographic materials were produced exclusively by the authors using public-domain spatial data provided by the Brazilian Institute of Geography and Statistics (IBGE). Spatial analyses and map visualizations were performed using QGIS version 3.34. Geographic layers representing population distribution and territorial boundaries were derived from IBGE shapefiles, and no copyrighted, proprietary, or third-party basemap or satellite imagery was used in the map construction, allowing integrated visualization of communities, health units. Geographic accessibility was classified into three levels (high, medium, and low), based on linear distance from each community to the urban area of Santarém.

Data analysis included the construction of health service utilization indicators, such as the proportion of care received in the last year, proportion of care performed outside the community, care resolvability, predominant educational level, and average income. These indicators were integrated into comparative analyses between communities, characterizing territorial and organizational disparities. Statistical tests included chi-square, ANOVA, and Spearman correlation to explore associations between variables, as well as a disease prevalence heatmap produced in Python using the Seaborn library.

To complement the territorial analysis, the Composite Access Index (CAI) was developed to integrate geographic, service-related, and epidemiological dimensions of access to health care at the community level. The index was constructed using three core components: (i) geographic accessibility, measured by the linear distance (km) from each community to the urban center of Santarém; (ii) health service utilization indicators, including the proportion of individuals who reported having accessed health care in the previous year and the proportion of care received outside the community; and (iii) epidemiological burden, represented by the prevalence of multimorbidity within each community.

All variables were initially calculated at the community level and subsequently normalized using min–max scaling to a 0–1 range, according to the formula: $X' = (X - X_{min})/ (X_{max} - X_{min})$, where higher values indicate poorer access conditions. After normalization, variables were combined using an equal-weight additive approach, in which each component contributed proportionally to the final index score. The CAI was therefore calculated as the arithmetic mean of the standardized components. The final index ranges from 0 to 1, with higher values representing greater barriers to access and

higher territorial vulnerability. To assess the robustness of the Composite Access Index, an exploratory sensitivity analysis was performed by applying alternative weighting schemes, including increased weights for the geographic component and for epidemiological burden. These alternative specifications produced minimal changes in the relative ranking of communities, indicating that the CAI is stable across reasonable variations in weighting assumptions.

Additionally, a Random Forest classification model was applied to identify health conditions most strongly associated with multimorbidity. The outcome variable was defined as the presence of multimorbidity (≥2 self-reported conditions), coded as a binary variable. Predictor variables included the set of self-reported diseases assessed in the questionnaire. Model training and validation were performed using a random split of the dataset into training (70%) and testing (30%) subsets. Class imbalance was assessed prior to modeling and addressed through class-weight adjustment within the algorithm to reduce bias toward the majority class. Missing data were minimal and handled through complete-case analysis.

Hyperparameters were tuned using grid search, with the number of trees, maximum tree depth, and minimum samples per node selected based on model performance in the training set. Model performance was evaluated in the test set using multiple metrics, including overall accuracy, sensitivity, specificity, and the area under the receiver operating characteristic curve (AUC). Variable importance was estimated using the mean decrease in impurity, providing a ranking of predictors based on their contribution to classification performance.

The study followed the ethical standards established by Resolution CNS No. 466/2012 and complied with the principles of the Declaration of Helsinki. Participant recruitment occurred between September 1, 2024, and October 30, 2025. All individuals invited to participate received detailed information about the study objectives and procedures, and written informed consent was obtained prior to data collection. Signed consent forms were securely stored as part of the study documentation, ensuring verification of voluntary participation. The research protocol was approved by the Research Ethics Committee of the Universidade do Estado do Pará (UEPA), under opinion No. 4.915.684, guaranteeing confidentiality, autonomy, and the protection of participants' rights throughout all stages of the research.

## Results

Data from 518 residents belonging to the Quilombola communities of Santarém-PA were analyzed, focusing on the assessment of territorial and organizational accessibility to health services and the identification of epidemiological patterns associated with the risk of multimorbidity. The results are presented regarding the geographic classification of accessibility, indicators of service use, resolvability, integrated territorial risk, and predictive modeling.

Table 1 presents the criteria adopted for stratifying geographic accessibility based on the linear distance between the communities and the urban area of Santarém. This classification defines three levels: high, medium, and low accessibility, based on distances traveled, estimated travel time, and epidemiological implications.

Communities located up to 5 km from the urban zone were classified with high accessibility, suggesting a lower risk of physical barriers to regular access to health services. Distances greater than 5 km and up to 25 km comprised the medium accessibility category, indicating dependence on transportation and greater susceptibility to delays. Communities situated more than 25 km away make up the low accessibility category, where there is a marked probability of care discontinuity, especially during periods of river seasonality.

Table 1. Criteria for classifying geographic accessibility based on distance and travel time (n = 9 communities).

| Classification | Distance (km) | Estimated Travel Time | Epidemiological Interpretation |
|---|---|---|---|
| High Accessibility | 0–5 km | ≤ 30 minutes | Favorable conditions for regular access; lower risk of territorial barriers. |
| Medium Accessibility | >5–25 km | 30–90 minutes | Moderate access, subject to logistical limitations, dependent on available transport. |
| Low Accessibility | > 25 km | > 90 minutes | Difficult access; higher risk of delays, abandonment, and care discontinuity. |

Table 2 synthesizes accessibility to PHC services, considering distance, the presence of Basic Health Units (UBS) in the territory, and qualitative barrier analysis. It is noted that: Pérola do Maicá, located within the high accessibility radius, has greater ease of travel and better care continuity. Arapemã and Saracura, despite their proximity (6.5 and 9.9 km, respectively, in a straight line), are accessed by river transport, which is not regular and does not follow a straight line due to necessary detours for better navigability. Bom Jardim, João Pereira, and Murumurutuba show intermediate accessibility, marked by dependence on river or land transport, with direct impacts on response time and care regularity.

Murumuru and Ituqui represent scenarios of low accessibility, with distances exceeding 25 km and strong vulnerability associated with transport conditions. Tiningu, despite being classified in the Table as High due to having its own UBS, is located 40 km away, which characterizes low geographic accessibility, although with partial mitigation at the primary level, indicating that the presence of a UBS in the community does not eliminate territorial barriers, especially when there is great distance from diagnostic support services and medical referrals.

As shown in Table 3, most medium complexity services, such as the Elderly CRS, Child CRS, Women's CRS, and the "Melhor em Casa" Program, are concentrated in the urban perimeter of Santarém. Thus: Communities with high geographic accessibility (Pérola do Maicá) have a structural advantage in accessing these services. Communities with medium and low accessibility face significant logistical barriers, such as long travel times, dependence on boats, and indirect costs, which can compromise the monitoring of chronic diseases and therapeutic continuity. This pattern reflects the urban centralization of specialized care provision and confirms inequalities in longitudinal care.

Table 2. Accessibility to Primary Health Care (PHC) services in quilombola communities of Santarém-PA, considering distance, service availability, and territorial barriers (n = 9 communities).

| Community | Distance from Urban Area (km) | Classification | Available PHC Service | Accessibility Analysis |
|---|---|---|---|---|
| Pérola do Maicá | 1.5 | High | UBS Maicá | Best access condition; proximity favors care continuity. |
| Arapemã | 6.5 | Medium | UBS Tapará | Dependent on river travel. |
| Saracura | 9.9 | Medium | UBS Tapará | Dependent on river travel. |
| Bom Jardim | 16.0 | Medium | UBS Jacamim | Limited access; dependence on river/road transport. |
| João Pereira | 20.0 | Medium | UBS Jacamim | Limited access; dependence on river/road transport. |
| Murumurutuba | 23.5 | Medium | UBS Tiningú | Upper limit of medium accessibility; significantly longer travel. |
| Murumuru | 25.0 | Low | UBS Tiningú | Difficult access, especially during flood/dry seasons. |
| Tiningu | 40.0 | High* | UBS Tiningú (own) | Has its own UBS (but geographically distant). |
| Ituqui | 30.0 | Low | UBS Tapará Miri (region) | Difficult access; high territorial vulnerability. |

Table 3. Accessibility to medium-complexity health services among quilombola communities of Santarém-PA (n = 9 communities).

| Service | Communities with High Accessibility | Communities with Medium/Low Accessibility |
|---|---|---|
| Elderly CRS | Pérola do Maicá | Arapemã, Saracura, Bom Jardim, João Pereira, Murumurutuba, Murumuru, Tiningu, Ituqui |
| Child CRS | Pérola do Maicá | Arapemã, Saracura, Bom Jardim, João Pereira, Murumurutuba, Murumuru, Tiningu, Ituqui |
| Women's CRS | Pérola do Maicá | Arapemã, Saracura, Bom Jardim, João Pereira, Murumurutuba, Murumuru, Tiningu, Ituqui |
| Melhor em Casa Program | Pérola do Maicá | Arapemã, Saracura, Bom Jardim, João Pereira, Murumurutuba, Murumuru, Tiningu, Ituqui |

Table 4 highlights that, although services such as SAMU 192 (Mobile Emergency Service) are available to all communities, this is only at the level of activation. Structural barriers remain, especially related to response time in critical situations. For distant communities, the estimated time exceeds 60 minutes and can be worsened by river seasonality. In the field of mental health and infectious diseases, services such as CAPS AD III, CAPSi, and CTA/SAE have viable accessibility only for communities close to the urban area. Communities located more than 20 km away face substantial difficulties, with a direct impact on prevention, diagnosis, and timely treatment.

Table 5 presents the synthesis of accessibility classification considering PHC, medium, and high complexity. The communities were then categorized into four risk levels: Low risk: Pérola do Maicá, which has better infrastructure, shorter distance, and greater resolvability. Moderate risk: Arapemã, Saracura, Bom Jardim, and João Pereira, which depend on intermediate travel. Moderate/High risk: Murumurutuba, due to distance and transport fragility. High risk: Murumuru and Tiningu, with geographic barriers and dependence on external services. Very high risk: Ituqui, which concentrates the highest vulnerabilities. Communities with a distance greater than 25 km show high or very high risk, regardless of the presence of a UBS. This pattern confirms the influence of territorial inequalities on Quilombola health vulnerability.

Fig 1 presents the spatial distribution, evidencing an unequal territorial pattern of accessibility to health services. A high concentration of health facilities is observed in the urban and peri-urban area of Santarém, especially near the Pérola do Maicá community, which is located practically within the urban fabric and therefore presents high geographic accessibility. This proximity results in greater service availability, shorter travel time, and greater care resolvability.

On the other hand, communities situated on the right bank of the Amazon River, such as Arapemã and Saracura, although geographically more distant from the urban area and accessed by intermittent river transport, present an intermediate pattern of connectivity and tend to access urban services with lower time cost. In contrast, Bom Jardim, João

**Table 4. Access to high-complexity and emergency health services in quilombola communities of Santarém-PA and main identified barriers (n = 9 communities).**

| Service | Type | Communities with Viable Access | Main Identified Barriers |
|---|---|---|---|
| SAMU 192 | Emergency | All communities (via telephone) | Response time >60 min for distances >25 km; difficulty locating; river seasonality. |
| CEREST | Occupational Health | Pérola do Maicá | Distance and transport compromise access for more remote communities. |
| CAPS AD III | Mental Health | Pérola do Maicá | Access unviable for distant communities without regular transport. |
| CAPSi | Child Mental Health | Pérola do Maicá | Need for an accompanying adult limits adherence; prolonged travel. |
| CTA/SAE | STI/AIDS and Viral Hepatitis | Pérola do Maicá | Intense geographic barrier for communities >20 km; risk of late diagnosis. |

**Table 5. Integrated territorial accessibility and health risk classification of quilombola communities of Santarém-PA (n = 9 communities).**

| Community | Distance from Urban Area (km) | PHC | Medium Complexity | High Complexity | Integrated Territorial Risk |
|---|---|---|---|---|---|
| Pérola do Maicá | 1.5 | High | High | High | Low |
| Arapemã | 6.5 | High | Medium | Medium | Moderate |
| Saracura | 9.9 | High | Medium | Medium | Moderate |
| Bom Jardim | 16 | Medium | Low | Low | Moderate |
| João Pereira | 20 | Medium | Low | Low | Moderate |
| Murumurutuba | 23.5 | Medium | Low | Low | Moderate/High |
| Murumuru | 25 | Low | Low | Low | High |
| Tiningu | 40 | High* | Low | Low | High |
| Ituqui | 30 | Low | Low | Low | Very High |

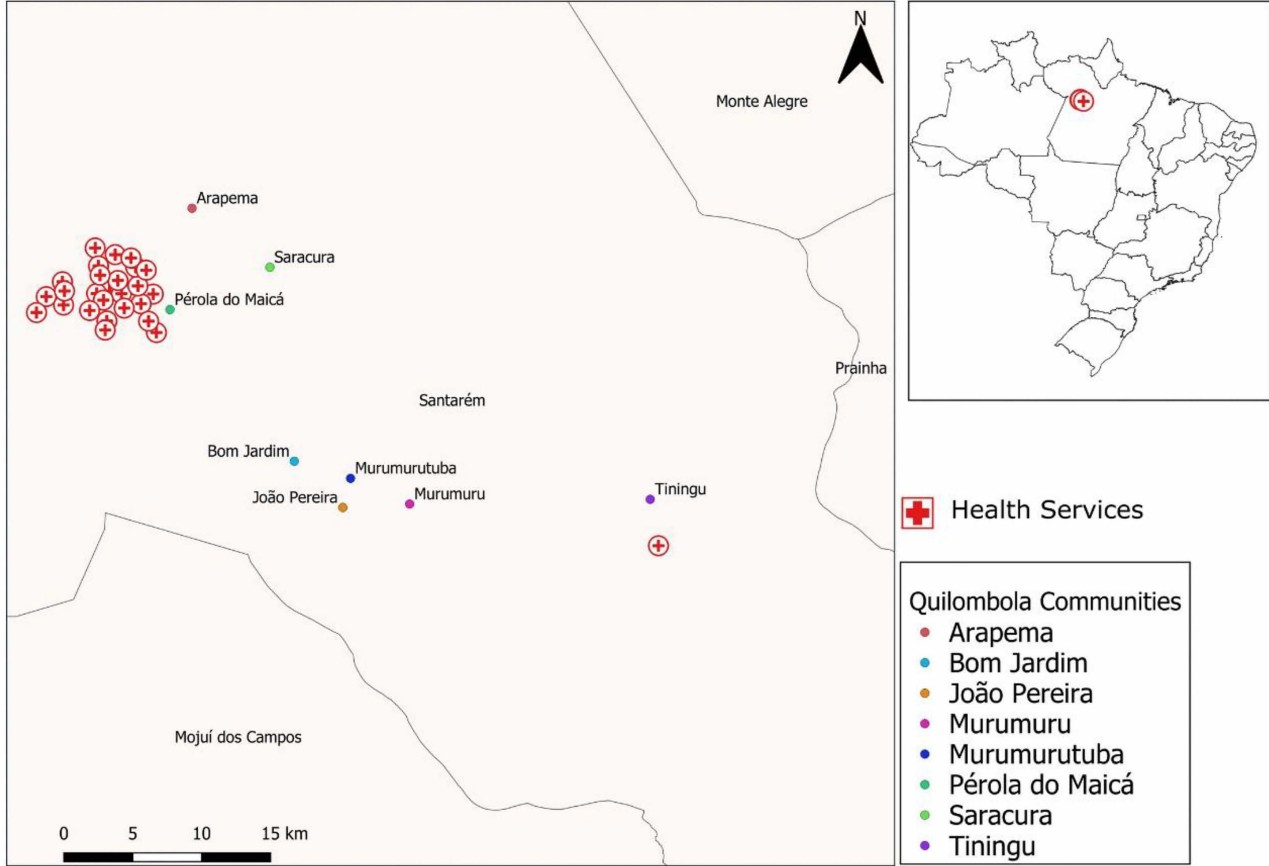

**Fig 1. Geographic distribution of quilombola communities and health services in the municipality of Santarém-PA, Lower Amazon region (n = 9 communities).** Source: Map created by the authors using public-domain shapefiles from the Brazilian Institute of Geography and Statistics (IBGE). The map was produced in QGIS software without the use of proprietary basemaps or satellite imagery, available at the link https://www.ibge.gov.br/geociencias/downloads-geociencias.html?caminho=cartas_e_mapas/mapas_municipais/colecao_de_mapas_municipais/2020/PA/.

Pereira, Murumurutuba, and Murumuru are located in areas with lower service density and greater dependence on land and river routes, often influenced by flood and dry seasons. This spatial distribution reinforces the classification observed in the indicators: these communities present medium to low accessibility, which is associated with greater territorial vulnerability.

The Tiningu community, located approximately 40 km from the urban center, is the most evident example of geographic isolation. Despite having its own UBS, its distance from medium and high complexity services represents a significant barrier. The spatial pattern of Tiningu reflects high geographic vulnerability, which contributes to a higher risk of care discontinuity and fewer opportunities to use specialized services (Fig 1).

Table 6 presents data on service utilization, location of care, and resolvability. The results show that: The highest rates of utilization in the last year occurred in Pérola do Maicá (95.0%), Tiningu (85.7%), and João Pereira (82.6%). The percentage of care received outside the community was high in all locations (70–95%), highlighting the centralization of services in the urban area. Resolvability showed a high proportion of completely resolved problems in some communities, such as Arapemã (100%), but was low in others, such as Pérola do Maicá (57.1%), suggesting that greater utilization does not necessarily translate into higher quality of care.

**Table 6. Health service access, utilization, and resolvability indicators among quilombola communities of Santarém-PA (total sample n = 518 participants).**

| Community | Distance (km) | Care Last Year | Outside Community | Problem Resolved | Predominant Education Level | Average Income |
|-----------|---------------|----------------|-------------------|------------------|-----------------------------|----------------|
| Arapemã | 6.5 | 42.9% | 66.7% | 100.0% | Elementary | 3.25 |
| Bom Jardim | 16.0 | 62.5% | 75.0% | 83.3% | Elementary | 2.83 |
| João Pereira | 20.0 | 82.6% | 83.7% | 85.7% | Elementary | 2.96 |
| Murumuru | 25.0 | 81.8% | 70.7% | 78.6% | Elementary | 3.33 |
| Murumurutuba | 23.5 | 64.3% | 78.6% | 80.0% | Elementary | 3.75 |
| Pérola do Maicá | 1.5 | 95.0% | 95.2% | 57.1% | High School | 3.33 |
| Saracura | 9.9 | 72.0% | 76.9% | 70.0% | High School | 2.50 |
| Tiningu | 40.0 | 85.7% | 72.7% | 72.7% | Elementary | 3.80 |
| Ituqui | 30.0 | N/A | N/A | N/A | Elementary | 2.70 |

Of the 518 participants included in the study, 273 reported having accessed health services in the previous year and were therefore eligible for the analysis of care resolvability. Table 7 shows that 72.5% of participants reported their problem was completely resolved, while 15.4% reported partial resolution. Only 4.0% reported that their demand was not met, and 8.1% did not know or did not respond. These results show a relatively satisfactory care performance, but still with heterogeneity among communities and a need to improve the quality of care provided.

Table 8 presents the CAI, constructed from the integration of distance, disease burden, and social indicators. The results show that: Ituqui (0.550), Tiningu (0.480), and Murumurutuba (0.331) presented the highest values, reflecting greater territorial vulnerability. Pérola do Maicá (0.076) presented the best index, compatible with its proximity to the urban zone. The CAI reinforces structural inequalities and supports the prioritization of differentiated strategies among territories.

The Random Forest modeling (Table 9) had excellent performance, with overall accuracy of 93.6%, allowing for the identification of determining factors for the occurrence of multimorbidity. The variables with the greatest importance were: Hypertension, Diabetes, Spinal disease, Arthritis/Rheumatism, Heart disease. These results corroborate the literature on chronic conditions as the central axis of the epidemiological profile of vulnerable populations.

The heatmap, Fig 2, highlights distinct patterns in the distribution of chronic and infectious diseases among the Quilombola communities. Greater density is observed in João Pereira and Murumurutuba, with intermediate prevalences of diabetes, spinal diseases, and arthritis/rheumatism, suggesting a more homogeneously distributed chronic morbidity profile in these communities. The presence of lighter shades in Tiningu for conditions like "Not applicable" indicates a high proportion of unreported diagnoses, which may point to low detection, underdiagnosis, or lower frequency of visits to services providing formal diagnosis. This pattern is consistent with the high geographic distance and the territorial risk classification attributed to this community.

Communities near the urban center, such as Pérola do Maicá, Arapemã, and Saracura, show more dispersed patterns, with low to moderate prevalences across various conditions, indicating more regular access and more distributed diagnosis. In Saracura, greater intensity in conditions like spinal disease and asthma stands out, suggesting a profile potentially associated with occupational exposure, physical strain, and environmental conditions.

Infectious diseases, such as malaria, dengue, and hepatitis, appear sporadically and with lower intensity, reflecting reduced prevalences in the communities, but with variations that suggest differentiated ecological exposure, especially in Murumuru and Murumurutuba, where there are moderate records of endemic Amazonian diseases.

## Discussion

The findings reveal a scenario that aligns closely with what various authors have described regarding the structural inequalities that shape the lives of quilombola populations. Understanding these inequalities becomes stronger when

**Table 7. Classification of health care resolvability among participants who accessed health services in the previous 12 months (n = 273).**

| Status | n | % |
|---|---|---|
| Problem completely resolved | 198 | 72.5% |
| Problem partially resolved | 42 | 15.4% |
| Problem not resolved | 11 | 4.0% |
| Not informed/ indeterminate | 22 | 8.1% |
| Total | 273 | 100.0% |

**Note:** This analysis includes only participants who reported having accessed health services in the previous 12 months (n = 273). Participants without reported service use during this period were not included in the resolvability analysis.

**Table 8. Composite Access Index (CAI) scores for quilombola communities of Santarém-PA, with higher values indicating poorer access to health services (n = 9 communities).**

| Community | Average CAI |
|---|---|
| Ituqui | 0.550 |
| Tiningu | 0.480 |
| Murumurutuba | 0.331 |
| Murumuru | 0.324 |
| João Pereira | 0.275 |
| Bom Jardim | 0.253 |
| Saracura | 0.191 |
| Arapemã | 0.139 |
| Pérola do Maicá | 0.076 |

**Table 9. Variable importance derived from the Random Forest model predicting multimorbidity among quilombola participants (model developed using data from n = 518 participants).**

| Variable | Importance |
|---|---|
| Hypertension | 0.192 |
| Diabetes | 0.111 |
| Spinal Disease | 0.088 |
| Arthritis/Rheumatism | 0.072 |
| Heart Disease | 0.067 |

analyzed through theoretical models addressing the social determination of health, particularly those that describe how environmental, socioeconomic, and political conditions shape access to services and influence patterns of illness within communities. In this regard, the framework proposed by Dahlgren and Whitehead stands out by illustrating successive layers of determination from individual factors to broader social structures, thus helping interpret the heterogeneity observed among the communities analyzed [10].

Given the cross-sectional design of the study, the findings should be interpreted as associations rather than causal relationships. Although geographic, organizational, and social barriers to health care access are strongly associated with patterns of multimorbidity and service use, the temporal direction of these relationships cannot be established. The discussion therefore emphasizes contextual and structural correlations, consistent with the study design.

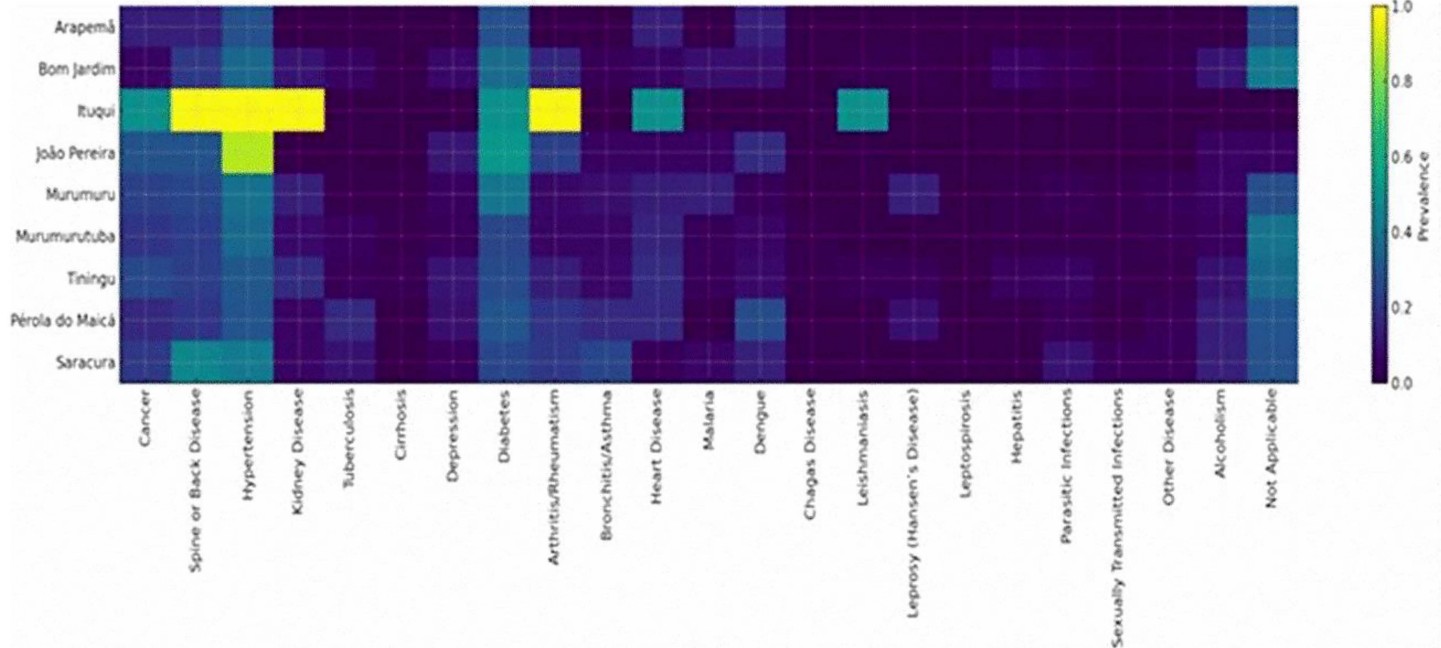

**Fig 2. Heatmap of self-reported disease prevalence by quilombola community in Santarém-PA (n = 518 participants).** Note: Color intensity represents disease prevalence, with darker shades indicating higher prevalence.

The set of geographic, economic, and organizational barriers identified in the quilombola territory also echoes the perspective of the National Commission on Social Determinants of Health, which considers access as a phenomenon conditioned by multiple interdependent elements rather than by the mere physical presence of services [11]. Analyzing access from this standpoint reveals that communities farther from urban centers tend to accumulate obstacles that extend beyond the territorial dimension, including indirect costs, dependence on river transportation, logistical difficulties, and reduced availability of specialized care.

In light of Michael Marmot's theory of health inequities, these findings reinforce the understanding that territorial differences are not random but reflect historically produced social structures that result in unequal opportunities and living conditions [2,12]. The unequal distribution of health services, particularly those of medium and high complexity, is associated with patterns in which more vulnerable populations experience greater difficulty accessing care, lower continuity of care, and a higher prevalence of adverse health outcomes. This interpretation positions geography as an inherent component of inequity rather than a mere environmental factor.

The literature on quilombola health demonstrates that vulnerabilities manifest not only through territorial restrictions but also through the organization of services and the degree to which institutional systems recognize their needs [5,13]. Studies across different regions highlight that geographic distance, though important, represents only one facet of the issue, as it interacts with income, education, transportation availability, environmental conditions, and cultural barriers to shape real experiences of accessing the SUS [14,15]. This pattern is confirmed in Santarém, where more isolated communities accumulate multiple layers of disadvantage.

Additionally, the World Health Organization recognizes that unfavorable socioeconomic conditions are associated with higher disease burden and reduced ability to seek timely care [16]. Among quilombola populations, such factors include limited access to income and education as well as historically rooted racial vulnerabilities, as discussed by Bailey et al. in

their analysis of structural racism and health [17]. This theoretical framework helps contextualize the multimorbidity patterns observed, particularly in communities with limited accessibility.

Research on quilombola health also indicates that traditional care practices, communication challenges with health professionals, and previous experiences of discrimination can influence service use and trust in the health system [18,19]. In Santarém, findings related to resolvability show that higher utilization does not necessarily translate into improved care experiences, suggesting possible weaknesses in care continuity, effectiveness, and patient reception.

The role of nursing in these territories also deserves emphasis. Studies indicate that professionals frequently face resource shortages, work overload, and logistical limitations, especially in rural and riverine areas [20–22]. Such conditions affect both the quality of care and the ability to perform surveillance, prevent diseases, and monitor chronic conditions. These limitations tend to be more pronounced in remote communities.

Another important aspect concerns emergency services in Amazonian territories. Although national policy expanded mobile emergency coverage through river ambulances (Ambulanchas), studies indicate persistent difficulties related to vessel maintenance, unsTable communication systems, and long response times [23,24]. Our findings align with these challenges, particularly in more isolated areas such as Tiningu and Ituqui, where residents often rely on private transport in critical situations.

From a public policy perspective, authors such as Freitas, Panno, and Almeida emphasize that the historical trajectory of quilombola communities in the Amazon has been marked by the formation of resistance territories that continue to face structural barriers to fully exercising constitutional rights, including health [25–27]. These challenges become evident when observing the distribution of health units, the centralization of specialized services in urban areas, and the need for long-distance travel for examinations and consultations.

Furthermore, health promotion as discussed by Durand and Heideman requires acknowledging sociocultural and environmental dimensions that shape ways of life and conditions of illness [28]. In quilombola communities, daily practices, forms of community organization, and support networks are essential to understanding both care-seeking behavior and decisions related to self-care.

When examining the patterns of multimorbidity and service utilization identified, it becomes evident that social, racial, and territorial inequalities intersect and are strongly associated with unequal health profiles among residents, as demonstrated in recent studies of quilombola regions [29–31]. The literature reinforces that Black and traditional populations face a higher burden of chronic diseases, often exacerbated by barriers to access, delayed diagnoses, and inconsistent follow-up.

The difficulty in accessing health information, highlighted by Norman and Skinner in their concept of digital health literacy, must also be considered [32]. In communities with limited connectivity, obtaining reliable information on prevention, disease management, and navigating the health system is challenging, directly influencing service use and self-care capacity.

Recent studies on therapeutic itineraries in the Amazon show that the path to care is shaped by complex decisions involving time, cost, climate, transportation availability, and expectations regarding care [18,33,34]. In the communities analyzed, these factors resonate strongly with findings related to problem resolution, place of first care, and reliance on services outside the community.

Taken together, the findings suggest that health access in quilombola communities is a multifaceted phenomenon shaped by overlapping structural, historical, social, cultural, and organizational conditions. Understanding how these layers interact is essential for guiding territorially sensitive public policies capable of promoting greater equity in health.

The Random Forest analysis should be interpreted as a predictive and exploratory tool rather than as an explanatory or causal model. Variable importance scores indicate the relative contribution of specific conditions to the classification of multimorbidity within the dataset, reflecting predictive relevance rather than etiological effects. Consequently, the identified conditions should not be interpreted as independent risk factors or causal determinants, but as markers that co-occur

more frequently with multimorbidity in this population. Despite these limitations, the use of Random Forest allowed the identification of patterns of co-occurring conditions in a complex, high-dimensional dataset, complementing traditional statistical analyses and strengthening the overall interpretation of multimorbidity patterns.

The use of linear (Euclidean) distance as a proxy for geographic accessibility represents an important methodological limitation, particularly in riverine and rural Amazonian contexts. In these territories, actual access to health services is strongly influenced by fluvial routes, road conditions, transportation availability, and seasonal variability related to flood and dry periods, which may substantially increase travel time beyond what is captured by straight-line distance. Consequently, Euclidean distance may underestimate real displacement effort and time, especially for communities located along complex river networks or unpaved road systems.

Despite this limitation, linear distance was adopted due to its practicality, transparency, and comparability across communities, as well as the absence of reliable, standardized data on travel time and transportation routes for all territories included in the study. Importantly, distance was not interpreted in isolation but integrated into the Composite Access Index alongside service-use indicators and epidemiological burden, partially mitigating the risk of oversimplification. Within this framework, distance functions as a structural marker of territorial isolation rather than a precise measure of travel experience.

Future studies could enhance this approach by incorporating travel time estimates, fluvial and road network analyses, or seasonally adjusted accessibility models, which would allow a more accurate representation of mobility constraints in Amazonian territories. Such refinements would further strengthen the assessment of health service access and contribute to more territorially sensitive planning and policy development.

## Conclusion

This study highlights that access to health care in quilombola communities of the Amazon is not primarily constrained by the physical presence of services, but by a set of structural conditions that shape how, when, and whether care can be effectively reached and sustained. Territorial isolation, fragile transportation systems, and the historical concentration of diagnostic and specialized services in urban centers interact with longstanding social inequities, producing differentiated patterns of multimorbidity and uneven opportunities for continuity of care across communities.

Rather than reflecting a simple gradient of distance, the observed disparities point to the persistence of territorially embedded disadvantages that limit timely diagnosis, follow-up, and chronic disease management, particularly in more remote settings. These findings reinforce the notion that geographic space operates as a social determinant of health, mediating the effects of broader political, economic, and organizational processes on everyday access to care. From this perspective, inequities in health among quilombola populations should be understood as the outcome of cumulative and historically constructed constraints, rather than as isolated logistical challenges.

At the same time, the interpretation of these results requires caution. The reliance on self-reported morbidity may underestimate the true burden of disease in contexts marked by restricted access to diagnostic services, where underdiagnosis is structurally embedded. The cross-sectional design precludes causal inference, and the use of linear distance as a proxy for accessibility does not fully capture the complexity of riverine mobility, seasonal variability, and transportation instability that characterize the Amazonian territory. In addition, although the Random Forest analysis contributed to identifying patterns of co-occurring conditions, it does not account for cultural practices, traditional care strategies, or levels of trust in health institutions, which may substantially influence health-seeking behaviors.

Despite these limitations, the study provides a territorialized analytical framework that is highly relevant for public health planning in traditional and rural contexts. The results suggest that strategies focused exclusively on expanding service coverage are unlikely to reduce inequities unless they are accompanied by investments in transportation infrastructure, diagnostic capacity, communication systems, and the strengthening of primary health care with community participation. Future research integrating longitudinal designs, qualitative approaches, and accessibility measures based on travel time

and fluvial networks will be essential to deepen understanding of how health inequities are produced and sustained in Amazonian quilombola territories, and to support the development of more context-sensitive and equitable health policies.

## Acknowledgments

We would like to thank the partnership with the project "Connecting Knowledge in the Amazon: Integrating Quilombola Communities and Public Schools with Computing and Artificial Intelligence", approved through a public call by the National Council for Scientific and Technological Development (CNPq) and carried out by the Federal University of Western Pará (UFOPA) and Project "Weaving Networks of Health and Knowledge: Innovation and Tradition in the Amazon", approved through a public call by the National Council for Scientific and Technological Development (CNPq) and carried out by the University of State Pará (UEPA).

## Author contributions

**Conceptualization:** Leanna Silva Aquino, Livia de Aguiar Valentim.

**Formal analysis:** Leanna Silva Aquino, Veridiana Barreto do Nascimento, Livia de Aguiar Valentim.

**Funding acquisition:** Veridiana Barreto do Nascimento.

**Investigation:** Leanna Silva Aquino, Ellen Mara Fernandes da Silva, Victoria Valentim Aguiar, Cesar Ferreira Fernandes Filho, Sheyla Mara Silva de Oliveira, Tatiane Costa Quaresma, Valney Mara Gomes Conde, Nádia Vicência do Nascimento Martins, Marcos Manoel Honorato, Guilherme Augusto Barros Conde, Franciane de Paula Fernandes, Livia de Aguiar Valentim.

**Methodology:** Leanna Silva Aquino, Ellen Mara Fernandes da Silva, Victoria Valentim Aguiar, Cesar Ferreira Fernandes Filho, Sheyla Mara Silva de Oliveira, Tatiane Costa Quaresma, Valney Mara Gomes Conde, Nádia Vicência do Nascimento Martins, Marcos Manoel Honorato, Guilherme Augusto Barros Conde, Franciane de Paula Fernandes, Livia de Aguiar Valentim.

**Writing – review & editing:** Leanna Silva Aquino, Ellen Mara Fernandes da Silva, Victoria Valentim Aguiar, Cesar Ferreira Fernandes Filho, Sheyla Mara Silva de Oliveira, Tatiane Costa Quaresma, Valney Mara Gomes Conde, Nádia Vicência do Nascimento Martins, Marcos Manoel Honorato, Veridiana Barreto do Nascimento, Guilherme Augusto Barros Conde, Franciane de Paula Fernandes, Livia de Aguiar Valentim.

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
