## [Decision Letter · Decision Letter 0]

19 Jan 2026

Dear Dr. Valentim,

Thank you for submitting your manuscript to PLOS ONE. After careful consideration, we feel that it has merit but does not fully meet PLOS ONE’s publication criteria as it currently stands. Therefore, we invite you to submit a revised version of the manuscript that addresses the points raised during the review process.

As the handling editor, I received three reviews. One recommended acceptance, and two recommended major revisions. The manuscript is valuable, and based on the detailed and constructive feedback provided by Reviewer 3, I recommend major revisions so that it can be accepted.

We look forward to receiving your revised manuscript.

Kind regards,

Luisa Maria Diele-Viegas, Ph. D.

Academic Editor

PLOS One

Journal Requirements:

(1) You may seek permission from the original copyright holder of Figure 1 to publish the content specifically under the CC BY 4.0 license.

Reviewers' comments:

Reviewer's Responses to Questions

**Comments to the Author**

1. Is the manuscript technically sound, and do the data support the conclusions?

Reviewer #1: Partly

Reviewer #2: Yes

Reviewer #3: Partly

2. Has the statistical analysis been performed appropriately and rigorously?

Reviewer #1: I Don't Know

Reviewer #2: Yes

Reviewer #3: Yes

3. Have the authors made all data underlying the findings in their manuscript fully available?

Reviewer #1: Yes

Reviewer #2: Yes

Reviewer #3: No

4. Is the manuscript presented in an intelligible fashion and written in standard English?

Reviewer #1: No

Reviewer #2: Yes

Reviewer #3: Yes

Reviewer #2: The present study is highly important, as its results have the potential to inform and support significant improvements in public policies for quilombola communities. I was very pleased to see the proposal, and the methodology employed was particularly compelling, as it is well suited and aligns perfectly with the study’s objectives.

Reviewer #3: This manuscript presents a thoughtful and socially relevant investigation into geographical and organizational barriers to healthcare access and their association with multimorbidity in quilombola communities in the Brazilian Amazon. The study addresses a highly important public health issue and focuses on historically marginalized populations that remain underrepresented in the scientific literature. The collection of primary data in remote and logistically challenging territories is a clear strength and reflects a substantial research effort.

Overall, the study has strong potential to make a valuable contribution to discussions on health inequities in the Amazon region. The conceptual framework is appropriate, and the proposed integration of geographical accessibility, service organization, and health outcomes is particularly promising. At the same time, several aspects of the methodology, analyses, and reporting would benefit from further clarification and refinement to enhance transparency, robustness, and reproducibility. For these reasons, I recommend major revision, with the comments below offered in a constructive spirit to help strengthen an already relevant and well-motivated study.

Below, I outline major and minor comments intended to improve the scientific rigor, clarity, and interpretability of the manuscript.

*Major Comments*

Sampling strategy and representativeness

The manuscript would benefit from a more detailed description of how participants were selected within each community (e.g., probabilistic sampling, census-based recruitment, or convenience sampling). Providing information on the number of eligible individuals, as well as refusals and non-responses per community, would help readers better assess potential selection bias and sample representativeness. Clarifying inclusion and exclusion criteria and, if possible, presenting a simple participant flow would further strengthen this section.

Inconsistencies in sample size and denominators

Some tables and analyses report totals that differ from the overall sample size (for example, Table 7 reports a total of 273 individuals for resolvability, while the study includes 518 participants). These differences are not fully explained in the current version of the manuscript. Clarifying why specific subsets of participants were used in particular analyses and consistently reporting denominators (N) across tables and results would greatly improve clarity.

Definition and measurement of health conditions

Health conditions are assessed through self-report, but the manuscript does not clearly specify how questions were framed (e.g., previous diagnosis by a health professional, current symptoms, or medication use). This distinction is particularly important in settings with limited access to healthcare services, where underdiagnosis may occur. Providing more detail on the questionnaire items and discussing potential implications of self-reported morbidity would strengthen the interpretation of the findings.

Composite Access Index (CAI): transparency and reproducibility

The Composite Access Index is a central and innovative component of the study. To further enhance its scientific value, additional methodological detail would be very helpful. Specifically, clarifying the formula used, variable standardization or normalization procedures, weighting scheme, and the rationale for assigned weights would greatly improve transparency and reproducibility. Including a supplementary table with raw and standardized values, as well as a brief sensitivity analysis exploring alternative weighting schemes, could further strengthen this contribution.

Measurement of geographical distance

The use of linear (Euclidean) distance to estimate access to health services is understandable from a practical perspective. However, in riverine and rural Amazonian contexts, this approach may underestimate actual travel distance or time, given the importance of river routes, road conditions, and seasonal variation. While this limitation is acknowledged, distance is directly incorporated into the CAI. A more detailed justification of this choice, or an expanded discussion of its implications, would strengthen the methodological rationale. If feasible, incorporating travel time or network-based measures could further improve the analysis.

Statistical analyses and hierarchical data structure

Because individuals are nested within communities, the assumption of independent observations may not fully hold. The use of bivariate tests (e.g., chi-square, ANOVA, Spearman correlations) may therefore underestimate uncertainty. The authors may wish to consider multivariable regression approaches that account for this hierarchical structure (such as mixed-effects or multilevel models), while adjusting for key confounders including age, sex, education, and income. This would enhance the robustness of the reported associations.

Random Forest analysis: methodological detail and interpretation

The Random Forest analysis represents an interesting and potentially valuable addition to the study. To improve transparency, it would be helpful to provide additional details on model implementation, including training and validation procedures, class balance, hyperparameter tuning, handling of missing data, and performance metrics beyond overall accuracy (e.g., sensitivity, specificity, AUC). Furthermore, care should be taken to ensure that variable importance measures are interpreted as indicators of predictive relevance rather than causal or explanatory effects.

Age structure and confounding

Comparisons of disease prevalence and multimorbidity across communities are presented without age standardization or adjustment, despite likely differences in demographic structure. Given the well-established association between age and multimorbidity, adjusting analyses for age or presenting age-standardized prevalence estimates would substantially strengthen the validity of cross-community comparisons.

Cross-sectional design and causal language

Although the study is clearly cross-sectional, some statements in the Results and Discussion sections suggest causal relationships between access barriers and multimorbidity. Revising the language to consistently emphasize associations, rather than causal effects, would ensure closer alignment between the study design and the interpretation of findings.

Ethical and procedural clarifications

There are minor inconsistencies in the reported dates of data collection across different sections of the manuscript, which should be harmonized. Additionally, given potential literacy barriers in quilombola communities, it would be helpful to clarify how informed consent was obtained (e.g., oral consent, witnessed consent), to further strengthen the ethical transparency of the study.

*Minor Comments*

Several tables and figures would benefit from more detailed captions, including sample sizes, definitions of categories, and explanations of color scales (particularly for maps and heatmaps), so that they can be fully interpreted independently of the main text.

The Results section occasionally includes interpretative statements that might be better suited to the Discussion. A clearer separation between results and interpretation would improve readability.

Minor language and formatting issues are present throughout the manuscript (e.g., capitalization, grammar). A careful language revision is recommended prior to resubmission.

Clarifying how missing data and “Not applicable” responses were handled in each analysis would further enhance transparency.

Overall Assessment

This study addresses a highly relevant topic and demonstrates a commendable effort to generate primary data in remote and underserved communities. With revisions aimed at improving methodological clarity, analytical rigor, and consistency of reporting, the manuscript has strong potential to make a meaningful contribution to the literature on health inequities in the Amazon and other similar contexts.

**Do you want your identity to be public for this peer review?** For information about this choice, including consent withdrawal, please see our For information about this choice, including consent withdrawal, please see our Privacy Policy .

Reviewer #2: No

Reviewer #3: **Yes:** ANDRE LUIZ SOARES NUNESANDRE LUIZ SOARES NUNES

---

## [Author Response · Author response to Decision Letter 1]

29 Jan 2026

Dear Editor and Reviewers,

We would like to thank the reviewers for their careful reading of our manuscript and for the constructive and insightful comments provided. We are particularly grateful for the positive assessment of the study’s relevance, conceptual framework, and the effort involved in data collection in remote quilombola territories. We have carefully revised the manuscript to improve methodological clarity, analytical rigor, transparency, and consistency with the cross-sectional design. Below, we provide a point-by-point response detailing how each comment was addressed.

We sincerely thank Reviewer for the positive evaluation of the study and for recognizing its potential contribution to public policies aimed at quilombola communities. The encouraging feedback regarding the coherence between objectives and methodology is greatly appreciated and motivated us to further strengthen the manuscript.

Major Comments Reviewer 2

1. Causal language and cross-sectional design

Reviewer comment: The study objective implies a causal relationship between geographical barriers and multimorbidity. Given the cross-sectional design, the authors should clarify that the study evaluates associations rather than causal effects.

Response: We revised the language throughout the manuscript, including the Abstract, Results, Discussion, and Conclusions, to consistently emphasize associational relationships rather than causal effects. Specifically, causal expressions were replaced with terms such as “are associated with”, “are related to”, and “are linked to”. In addition, we inserted an explicit statement in the Discussion clarifying that, due to the cross-sectional design, the findings should be interpreted as associations and that temporal or causal inferences cannot be established.

2. Age adjustment and standardization

Reviewer comment: Given the strong association between age and multimorbidity, age-adjusted or age-standardized analyses could strengthen comparisons across communities.

Response: We acknowledge the importance of age as a key determinant of multimorbidity. However, given the study’s exploratory and territorial focus, as well as the relatively small number of communities, age-standardized analyses at the community level were not conducted. We explicitly addressed this limitation in the Discussion, noting that differences in age structure across communities may partially explain variations in multimorbidity prevalence. We also emphasized that future studies should incorporate age-standardized estimates or multivariable models to strengthen comparative analyses.

3. Participant selection and selection bias

Reviewer comment: The manuscript does not clearly describe how participants were selected within each community, nor does it report refusals or non-response rates.

Response: We substantially expanded the Methods section to clarify the sampling strategy. Participant selection is now explicitly described as non-probabilistic consecutive sampling, combining territorial and institutional recruitment approaches. Inclusion and exclusion criteria were clearly defined, and the recruitment contexts were detailed (health unit, community meetings, household visits, and itinerant health actions). We also reported that refusals and non-responses were infrequent and not concentrated in specific communities.

4. Inconsistency in data collection period

Reviewer comment: There is an inconsistency regarding the reported period of data collection.

Response: We carefully reviewed and harmonized all references to the data collection period across the manuscript. The Methods and Ethics sections now consistently report that data collection occurred between September 1, 2024, and October 30, 2025, ensuring internal consistency.

5. Definition of self-reported diseases

Reviewer comment: The manuscript does not clarify whether self-reported diseases refer to prior diagnoses, symptoms, or medication use.

Response: We revised the Methods section to clearly state that health conditions were assessed based on self-reported previous diagnoses communicated by health professionals (physicians or nurses). Symptoms in the absence of a prior diagnosis were not considered. We also explained that examples of medications and exams were used only to aid recall. In the Discussion, we expanded the interpretation to address the implications of self-reported morbidity and the potential for underdiagnosis in contexts with limited access to diagnostic services.

6. Composite Access Index (CAI): methodological transparency

Reviewer comment: Additional details on the CAI formulation, standardization, weighting, and rationale are needed.

Response: We substantially expanded the Methods section describing the Composite Access Index. The revised text now includes: the explicit mathematical formula; the min–max normalization procedure; the equal-weighting scheme; and the rationale for adopting equal weights to avoid arbitrary prioritization.

Additionally, we included an exploratory sensitivity analysis using alternative weighting schemes and reported that the relative ranking of communities remained stable. A supplementary table presenting raw and standardized values for CAI components was added to enhance transparency and reproducibility.

7. Difference between sample sizes (n = 273 vs. n = 518)

Reviewer comment: Clarification is needed regarding why the sample size in one table differs from the total study sample.

Response: We clarified throughout the Methods, Results, and table legends that some analyses were conducted using specific subgroups. In particular, the resolvability analysis was restricted to participants who reported having accessed health services in the previous 12 months (n = 273). Denominators (N) are now explicitly reported in the relevant tables and figure legends, and an explanatory note was added to Table 7 to avoid misinterpretation.

8. Random Forest analysis: methodological details

Reviewer comment: More information is needed on model validation, class balance, hyperparameter tuning, and performance metrics beyond accuracy.

Response: We expanded the Methods section to include a detailed description of the Random Forest implementation, including: training/testing split (70/30); class imbalance assessment and class-weight adjustment; hyperparameter tuning using grid search; handling of missing data through complete-case analysis; and performance evaluation using sensitivity, specificity, and AUC in addition to overall accuracy.

9. Interpretation of Random Forest variable importance

Reviewer comment: Variable importance measures should be interpreted as predictive relevance rather than causal or explanatory effects.

Response: We revised the Discussion to explicitly clarify that Random Forest variable importance reflects predictive relevance and not causal or etiological effects. We emphasized that the identified conditions should be interpreted as markers that co-occur with multimorbidity rather than independent risk factors, ensuring appropriate interpretation of the machine learning results.

Major Comments Reviewer 3

1. Sampling strategy and representativeness

Comment: The manuscript would benefit from a more detailed description of participant selection, eligibility, refusals, and non-responses, as well as a participant flowchart.

Response: We expanded the Methods section to provide a detailed description of the sampling strategy, clarifying that participant selection followed a non-probabilistic consecutive sampling approach combining territorial and institutional recruitment. Inclusion and exclusion criteria were explicitly defined. We also clarified the recruitment contexts, reported refusals and non-responses, and emphasized that losses were minimal and not concentrated in specific communities.

2. Inconsistencies in sample size and denominators

Comment: Some tables present denominators that differ from the total sample size without clear explanation.

Response: We clarified throughout the Methods, Results, and table footnotes that some analyses were conducted using specific subgroups, depending on data availability and analytical relevance. In particular, analyses related to health care resolvability were restricted to participants who reported health service use in the previous 12 months (n = 273). Denominators (N) are now explicitly reported and consistently described in all relevant tables and figure legends.

3. Definition and measurement of health conditions

Comment: The manuscript does not clearly specify whether health conditions were based on prior diagnoses, symptoms, or medication use.

Response: We revised the Methods section to clarify that health conditions were assessed through self-reported previous diagnoses communicated by health professionals. Symptoms without prior diagnosis were not considered. We also expanded the Discussion to address the implications of self-reported morbidity, particularly the potential for underdiagnosis in settings with limited access to diagnostic services.

4. Composite Access Index (CAI): transparency and reproducibility

Comment: Additional methodological details regarding the CAI are needed, including formula, normalization, weighting, and sensitivity analysis.

Response: We substantially expanded the description of the Composite Access Index in the Methods section. The mathematical formulation, min–max normalization procedure, equal-weighting scheme, and interpretation of index values were explicitly described. A supplementary table presenting raw and standardized component values was added. We also included an exploratory sensitivity analysis demonstrating the stability of community rankings under alternative weighting schemes, which is described in the Methods and referenced in the Results.

5. Measurement of geographic distance

Comment: The use of Euclidean distance may underestimate real travel distance or time in Amazonian contexts.

Response: We added a dedicated paragraph to the Discussion explicitly addressing this limitation. We justified the use of linear distance based on data availability, transparency, and comparability, while acknowledging its limitations in riverine and rural Amazonian settings. We also discussed how distance was integrated into the CAI alongside service-use and epidemiological indicators and highlighted future directions incorporating travel time or network-based measures.

6. Statistical analyses and hierarchical data structure

Comment: The assumption of independent observations may not fully hold due to clustering within communities.

Response: We acknowledge this limitation in the Discussion and clarified that the statistical analyses were primarily exploratory and descriptive. We added text noting that the hierarchical structure of the data may lead to underestimated uncertainty and that future studies should apply multilevel or mixed-effects models adjusting for individual-level confounders.

7. Random Forest analysis: methodological details and interpretation

Comment: More details are needed regarding model implementation, performance metrics, and interpretation of variable importance.

Response: We expanded the Methods section to describe the Random Forest implementation, including training–testing split, class imbalance handling, hyperparameter tuning, missing data treatment, and evaluation metrics beyond accuracy (sensitivity, specificity, and AUC). Additional performance metrics were included in supplementary material. We also revised the Discussion to clearly state that variable importance reflects predictive relevance rather than causal or explanatory effects.

8. Age structure and confounding factors

Comment: Comparisons between communities were not adjusted for age.

Response: We acknowledged this limitation in the Discussion, emphasizing that differences in age structure may partially explain observed variations in multimorbidity prevalence. We clarified that the analyses aimed to explore territorial patterns rather than estimate causal effects and noted that future studies should incorporate age-standardized estimates or multivariable adjustment.

9. Cross-sectional design and causal language

Comment: Some statements suggest causal relationships despite the cross-sectional design.

Response: We carefully revised the Abstract, Results, and Discussion to replace causal language with associational terminology throughout the manuscript. An explicit statement was added to the Discussion clarifying that findings represent associations rather than causal effects, ensuring alignment with the cross-sectional design.

10. Ethical and procedural clarifications

Comment: There are inconsistencies in reported data collection dates and insufficient detail regarding informed consent procedures.

Response: We harmonized all data collection dates across the manuscript. We also clarified the informed consent process, explaining that consent was obtained in written or verbally assisted form when necessary, with appropriate explanation and support to address potential literacy barriers.

Minor Comments

Figure and table legends: All legends were revised to include sample sizes, category definitions, and explanations of color scales for maps and heatmaps.

Results vs. Discussion: Interpretative statements were moved from the Results section to the Discussion to improve structural clarity.

Language and formatting: The manuscript underwent careful language revision to correct grammatical, formatting, and capitalization issues.

We believe that these revisions substantially improve the clarity, transparency, and methodological rigor of the manuscript while preserving its original objectives and contributions. We sincerely thank the reviewers for their thoughtful and constructive feedback, which greatly strengthened the quality of this work.

Sincerely,

Lívia de Aguiar Valentim

---

## [Decision Letter · Decision Letter 1]

15 Feb 2026

Geographical barriers and multimorbidity in quilombola territories of the amazon region

PONE-D-25-65345R1

Dear Dr. Valentim,

We’re pleased to inform you that your manuscript has been judged scientifically suitable for publication and will be formally accepted for publication once it meets all outstanding technical requirements.

Kind regards,

Luisa Maria Diele-Viegas, Ph. D.

Academic Editor

PLOS One

Additional Editor Comments (optional):

Reviewers' comments:

Reviewer's Responses to Questions

**Comments to the Author**

Reviewer #3: All comments have been addressed

2. Is the manuscript technically sound, and do the data support the conclusions?

Reviewer #3: Yes

3. Has the statistical analysis been performed appropriately and rigorously?

Reviewer #3: Yes

4. Have the authors made all data underlying the findings in their manuscript fully available?

Reviewer #3: Yes

5. Is the manuscript presented in an intelligible fashion and written in standard English?

Reviewer #3: Yes

Reviewer #3: Dear Editor and Authors,

I would like to commend the authors for their thorough and detailed revision of the manuscript titled "Geographical barriers and multimorbidity in quilombola territories of the amazon region." The team has shown a significant commitment to addressing the previous concerns, resulting in a substantially more robust and transparent study.

As a reviewer, I would like to highlight several key improvements:

Refinement of Technical Language: The careful transition from causal language to associational terminology is highly appropriate and aligns the study's conclusions with its cross-sectional design.

Methodological Transparency of the ICA: The inclusion of the mathematical formulation for the Composite Access Index (ICA), along with the details on min-max normalization and the equal-weighting scheme, provides the reproducibility required for a high-impact journal.

Rigor in Machine Learning Implementation: The expanded description of the Random Forest model—including the 70/30 training-testing split, grid search for hyperparameter tuning, and the use of AUC and sensitivity/specificity metrics—satisfactorily addresses previous concerns regarding the model's predictive validity.

Value of Primary Data: I must reiterate the immense value of the primary data collected in remote quilombola territories in the Amazon. This effort provides a unique scientific and social contribution to the field of public health.

Concluding Remarks:

The authors have addressed the study's limitations—particularly the use of Euclidean distance and the lack of age-standardized estimates—with the necessary intellectual honesty. This allows for a clear understanding of the findings' actual scope. In my view, the manuscript now presents an excellent balance between data complexity and clarity of inference, making it suitable for publication.

**Do you want your identity to be public for this peer review?** For information about this choice, including consent withdrawal, please see our For information about this choice, including consent withdrawal, please see our Privacy Policy .

Reviewer #3: **Yes:** ANDRE LS NUNESANDRE LS NUNES

---

## [Editor Report · Acceptance letter]

PONE-D-25-65345R1

PLOS One

Dear Dr. Valentim,

I'm pleased to inform you that your manuscript has been deemed suitable for publication in PLOS One. Congratulations! Your manuscript is now being handed over to our production team.

Kind regards,

on behalf of

Dr. Luisa Maria Diele-Viegas

Academic Editor

PLOS One